# BHRF1 Enhances EBV Mediated Nasopharyngeal Carcinoma Tumorigenesis through Modulating Mitophagy Associated with Mitochondrial Membrane Permeabilization Transition

**DOI:** 10.3390/cells9051158

**Published:** 2020-05-07

**Authors:** Shujie Song, Zhiying Jiang, David Ethan Spezia-Lindner, Ting Liang, Chang Xu, Haifeng Wang, Ye Tian, Yidong Bai

**Affiliations:** 1School of Public Health, Xi’an Jiaotong University, Xi’an 710061, Shaanxi, China; ssj199042@sina.com; 2Key Laboratory of Laboratory Medicine, Ministry of Education, Zhejiang Provincial Key Laboratory of Medical Genetics, College of Laboratory Medicine and Life Sciences, Wenzhou Medical University, Wenzhou 325035, Zhejiang, China; jiangzhiying1993@sina.com (Z.J.); liangt@uthscsa.edu (T.L.); 3No. 3 Hospital, the Affiliated Hospital of Northwest University School of Medicine, Xi’an 710018, Shaanxi, China; xasyjykwhf@sina.com; 4Department of Cell Systems and Anatomy, University of Texas Health San Antonio, San Antonio, TX 78258, USA; spezialindnd@livemail.uthscsa.edu; 5First Affiliated Hospital of Wenzhou Medical University, Wenzhou 325000, Zhejiang, China; singx21@sina.com

**Keywords:** NPC, EBV, BHRF1, mitochondria, mitochondrial membrane permeabilization transition (MMPT), mitophagy

## Abstract

Epstein–Barr virus (EBV) is a major contributor to nasopharyngeal carcinoma (NPC) tumorigenesis. Mitochondria have been shown to be a target for tumor viral invasion, and to mediate viral tumorigenesis. In this study, we detected that mitochondrial morphological changes in tumor tissues of NPC patients infected with EBV were accompanied by an elevated expression of BHRF1, an EBV encoded protein homologue to Bcl-2. High expression of BHRF1 in human NPC cell lines enhanced tumorigenesis and metastasis features. With BHRF1 localized to mitochondria, its expression induced cyclophlin D dependent mitochondrial membrane permeabilization transition (MMPT). The MMPT further modulated mitochondrial function, increased ROS production and activated mitophagy, leading to enhanced tumorigenesis. Altogether, our results indicated that EBV-encoded BHRF1 plays an important role in NPC tumorigenesis through regulating cyclophlin D dependent MMPT.

## 1. Introduction

Nasopharyngeal carcinoma (NPC) is a squamous cell carcinoma that develops in the nasopharynx [1]. Epidemiologic study has implicated Epstein–Barr virus (EBV) infection, genetic and environmental factors in NPC tumorigenesis [2]. The presence of the EBV genome in NPC cells has been well established [3,4]. EBV has a double-stranded, circular 172 kb genome which encodes more than 90 proteins, and it establishes a life-long infection in the host cells [5]. Several EBV-encoded proteins have been implicated in NPC tumorigenesis, and interestingly, some of these molecules have also been found to regulate mitochondrial function or are the homologues of mitochondrial proteins. For example, the latent membrane protein-1 (LMP1) promotes myeloid-derived suppressor cells to expand in the tumor microenvironment by inducing extra-mitochondrial glycolysis in NPC [6]. In addition, the LMP2A-mediated Notch pathway enhances mitochondrial fission by elevating dynamin-related protein 1 (Drp1), which also promotes cellular migration [7]. BARF1 regulates caspase-dependent mitochondrial apoptosis [8]. The Bcl-2 analogs BHRF1 and BALF1 preferentially localize to mitochondria [9].

Mitochondria are ubiquitous organelles in eukaryotic cells whose primary role is to generate energy in the form of ATP through oxidative phosphorylation [10]. The mitochondrial electron transport chain is also a major source of reactive oxygen species (ROS), as some of the electrons passing to molecular oxygen are instead leaked out of the chain. In addition to their cytotoxic effects, ROS participate in the regulation of many cellular pathways [11]. Studies have also shown that mitochondria play a central role in cell homeostasis by regulating both cell death and cell survival [12].

Apoptosis is a highly conserved cell death pathway that regulates cell homeostasis. Apoptosis can be activated intrinsically through mitochondrial pathways [13]. Intrinsic apoptotic signaling pathways are tightly regulated by Bcl-2 family members, through their effects on mitochondrial function including mitochondrial membrane permeabilization transition (MMPT)[14]. On the other hand, autophagy is an evolutionarily conserved pathway which plays an essential role in maintaining cellular homeostasis through lysosomal degradation of complex biomolecules into components that can be recycled [15]. Autophagy is also an adaptive process that responds to metabolic stress [16]. Mitochondrial autophagy is termed “mitophagy” [17].

In this study, we set out to understand the role of mitochondria in EBV-mediated NPC tumorigenesis. We initiated our investigation by comparing nasopharyngeal cancer tissues and nasopharyngitis tissues obtained from 60 patients with nasopharyngeal disease. We then further studied the mechanism with molecular and biochemical experiments in two EBV-negative human NPC cell lines, namely CNE1 and 5-8F. Some of these molecular results were then verified in NPC patient samples.

## 2. Materials and Methods

### 2.1. Tissues

Nasopharyngeal cancer tissues and nasopharyngitis tissues were obtained from 60 patients with nasopharyngeal disease who underwent endoscopy at The First Affiliated Hospital of Wenzhou Medical University from 10 April 2015 to 21 September 2016. All subjects gave their informed consent for inclusion before they participated in the study. The study was conducted in accordance with the Declaration of Helsinki, and the protocol was approved by the Ethics Committee of Wenzhou Medical University (2014006).

### 2.2. Cell Lines

CNE1 and 5-8F are EBV negative, human NPC cell lines. C666-1 is an EBV positive, human NPC cell line. CNE1 and 5-8F cell lines were obtained from Dr. Meng (Taizhou hospital, Taizhou, China), and the C666-1 cell line was provided by Anburui Biological Technology Co., Ltd. (Fujian, China). CNE1-BHRF1 and 5-8F-BHRF1 are NPC cell lines that stably express BHRF1. All cell lines were grown in DMEM (Dulbecco’s Modification of Eagle’s Medium) with high glucose (DMEM, HyClone, Logan, UT, USA) supplemented with 10% fetal bovine serum plus antibiotics. All cells were cultured in a humidified incubator maintaining 5% CO_2_ and 37 °C.

### 2.3. Construction and Identification of a Plasmid Containing BHRF1 Gene

According to the sequence information in Gene Bank (NCBI: NC_007605), we designed the primers 5′-CACTAGTCCAGTGTGGTGGCCACCATGGCCTATTCAACAAGGGAGATA-3′ (forward) and 5′-TGCTGGATATCTGCAGTGTCTTCCTCTGGAGATAAA-3′ (reverse) to clone the BHRF1 gene from the B95-8 cell line by PCR. The recombinant vector PCDH-BHRF1-flag was created and verified transformed by the Shanghai Bio-Engineering Company.

To construct a stable-transfected cell line that expresses BHRF1 protein, the 293T as the prepared cells, was transfected with packaging plasmid (PSPA, PMD2G) and PCDH-BHRF1-flag plasmid using lipofectamine-3000 reagent, establishing the BHRF1 virions. 24 h later, the filtered virus was harvested with a 0.45 μm filter. When the CNE1 and 5-8F cells achieved 40–50% confluence in 6-well plates, the cell culture medium was changed to 1 mL DMEM high glucose medium and 1 mL BHRF1 virus. After 10 h, the medium was replaced with medium supplemented with 10% FBS and incubated for a further 48 h in 5% CO_2_ at 37 °C. Finally, the cells were screened for the BHRF1 gene with 0.9 μg/mL puromycin for about 10 days and the clonal cell clusters were verified using a western blotting assay.

### 2.4. Mitochondrial Localization

Cells were suspended in TD buffer (W), and were homogenized on ice until 80% of the cells were stained blue by trypan blue under a microscope. The lysates were centrifuged at 12,000× *g* at 4 °C for 3 min, and then the supernatant was transferred to a new Eppendorf tube and subjected to a centrifugation again under the same conditions. Then, the supernatant was collected and centrifuged at 15,000× *g* for 2 min at 4 °C. The freshly extracted mitochondria (M) were suspended with 60 μL TD buffer and centrifuged. After removing the supernatant, the sediment underwent lysis with 30 μL 0.1 mM Na_2_CO_3_ for 30 min on ice and centrifugation at 75,000× *g* at 4 °C for 40 min. The reaction mixtures were centrifuged at 170,000× *g* at 4 °C for 30 min to separate the precipitate (M2, mitochondrial membrane) and supernatant (M1, mitochondrial matrix) fractions. The fractions were subjected to SDS-PAGE followed by Western blots.

For separating the inner and outer membrane of mitochondria, Proteinase K (Pro K) was used to digest the mitochondrial outer membrane, Pro K with Triton × 100 (20%) was used to digest the inner and outer mitochondrial membrane, and only Triton × 100 (20%) was used to digest neither. Then, 3.33 μL 1 mg/mL Pro K was added to the 30 μL freshly extracted mitochondria for 1 h and then mixed with 1 μL PMSF to obtain the mitochondrial inner membrane. A mixture of 3.33 μL 1 mg/mL Pro K and 3.33 μL Triton × 100 (20%) was added to the 30 μL freshly extracted mitochondria for 1 h and then mixed with 1 μL PMSF, which yielded nothing. Subsequently, 3.33 μL Triton × 100 (20%) alone was added into the 30 μL freshly extracted mitochondria for 1 h and then mixed with 1 μL PMSF in order to harvest everything. The reaction mixtures were separated by SDS-PAGE and analyzed by Western blots.

### 2.5. Colony Formation Assay

The cells were diluted to 4 × 10^2^ cells/mL in six-well plates, seeding 2 mL cell suspensions per well and incubating with 5% CO_2_ at 37 °C for 7–10 days. The cells were washed with PBS and fixed in 4% paraformaldehyde for 20 min, then were washed twice with PBS and stained with 1% crystallized purple dye for 30 min. After washing three times with PBS, the number and size of cell colonies were analyzed by a microscope.

### 2.6. CCK-8 Cell Viability Assay

Multiple identical samples of 1 × 10^4^ cells were placed on 96-well plates in 100 μL of DMEM and were cultured at 37 °C for 0, 24, 48, or 72 h. The medium was replaced the following day with 10% FBS DMEM medium with 10 μL CCK-8 for 1 h at 37 °C. Cell counting and viability analysis were performed using a microplate reader by a Varioskan™ Flash Multimode Reader (Thermo Scientific, Waltham, MA, USA).

### 2.7. Wound Closure Assay

The cells were diluted to 2 × 10^5^ cells/mL in 24-well plates, seeding 0.5 mL cell suspensions per well with 10% FBS DMEM medium, and then incubated in 5% CO_2_ at 37 °C for 1 day to prepare cells achieving 80% confluence. Then, a pin tool or needle was used to scratch and remove cells from a discrete area of the confluent monolayer to form a cell-free zone into which cells at the edges of the wound can migrate. After washing three times with PBS, cells were grown in DMEM medium with high glucose. Images of cell movement were captured at 0, 24, 48, and 72 h and the rate of wound closure was then calculated.

### 2.8. Transwell Invasion Experiment and Transwell Migration Assay

The 45 μL matrigel solution per transwell was prepared by combining 5 μL matrigel (melted at 4 °C) and 45 μL high glucose DMEM with serum-free medium and incubated in 5% CO_2_ at 37 °C for 12 h. Then, the transwells were put in 24-well plates with 600 μL 10% FBS DMEM medium. After growing in serum-free DMEM medium for 6 h, 1 × 10^3^ cells were resuspended in 200 μL of serum-free DMEM medium, then transferred to the prepared transwells and cultured in 5% CO_2_ at 37 °C for 48 h. The transwells were washed with PBS and fixed in 4% paraformaldehyde for 20 min, then were washed twice with PBS and stained with 0.1% crystallized purple dye for 20 min. The numbers of purple cells were examined by a microscope after eliminating covered cells under the transwell with PBS. The transwell migration assay was conducted in a similar way as the invasion assay, but without the matrigel solution.

### 2.9. Transmission Electron Microscopy Experiment

The prepared cells were fixed in 2.5% glutaraldehyde fixative (2.5% glutaraldehyde and 0.2 M HCl in 0.1 M cacodylate buffer, pH 7.2) for 2 h at 4 °C. Cells were fixed with 1% osmic acid for 1 h and stained with 1% uranium acetate for the same amount of time. Then, the cells were dehydrated in increasing concentrations of acetone (40%, 50%, 60%, 70%, 80%, 90%) for 15 min each and twice with 100% acetone for 20 min every time. Next, the samples were embedded in epoxy resin (TAAB 812 resin, TAAB Laboratories, Aldermaston, UK) at 45 °C for 6 h and 65 °C for 48 h. After resin embedding, ultra-thin sections were cut with an ultramicrotome using a diamond knife and collected onto 150 mesh copper grids (Electron Microscopy Sciences, Fort Washington, PA, USA). These sections were observed using a transmission electron microscope (H-800, Hitachi Ltd., Tokyo, Japan).

### 2.10. MMP Measurement

The cells were diluted to 4 × 10^4^ cells/mL in six-well plates, seeding 2 mL cell suspensions per well and were incubated in 5% CO_2_ at 37 °C for 2–3 days to prepare cells achieving 60–80% confluence. Cells were washed once with PBS and incubated with TMRE (tetramethylrhodamine, ethyl ester; Invitrogen), which was diluted with 10% FBS medium to 30 μM. We applied 1 mL dilute TMRE per well in 5% CO_2_ at 37 °C for 15 min and then cells were observed and photographed under a fluorescence microscope. To determine the membrane potential, we used fluorescence ratio (F-F0)/F0 to represent the potential, where F0 is the baseline fluorescence signal.

### 2.11. ROS Measurement

Mitochondrial ROS were measured according to our team’s published article [18]. In short, cells were washed in Hank’s buffered salt solution (HBSS), and resuspended in HBSS containing 5 µM Mito SOX (Molecular Probes, Carlsbad, CA, USA), culturing in 5% CO_2_ at 37 °C for 5 min. Then, cells were washed twice with HBSS and fluorescence was recorded by a Varioskan™ Flash Multimode Reader (Thermo Scientific, Waltham, MA, USA).

### 2.12. ATP Measurements

According to the manufacturer’s instructions, an ATP measurement kit (Molecular Probes, Carlsbad, CA, USA) was used as the main reagent to measure ATP. In short, the prepared cells achieving 60–80% confluence in 6-well plates were washed with PBS and then boiled in 100 μL boiling buffer (4 mM EDTA and 100 mM Tris, adjusted to pH 7.75 with acetic acid) at 100 °C for 90 seconds. Supernatants were retrieved via centrifugation at 10,000× *g* for 1 min. ATP content was determined by measuring the luminescence of supernatants mixed with luciferase assay buffer using a Varioskan™ Flash Multimode Reader (Thermo Scientific, Waltham, MA, USA). ATP luminescence was normalized via protein concentration.

### 2.13. Flow Cytometry

Fluorochrome-labeled Annexin V (Annexin V-FITC) was used to specifically target and identify early apoptotic cells. Late apoptotic cells, which have lost membrane integrity, are recognized by 7 -aminoactinomycin D (7-AAD). In contrast, necrotic cells lose membrane integrity early and become 7-AAD positive. The prepared cells were plated in 96-well plates at 1 × 10^6^ cells each, seeding 200 μL cell suspensions per well, and then treated or untreated with the concentrations as described in the text of CsA for 48 h with 5% CO_2_ at 37 °C. Cells were washed once with PBS and resuspended in 500 μL PBS. Then, 200 μL of cell suspensions were transferred into a tube and then incubated with 1.5 μL Annexin V and 1 μL 7-AAD (BD Biosciences, San Jose, CA, USA) for 30 min. After supernatants were discarded via centrifugation at 1500 rpm/min for 5 min, the cells were resuspended with 200 μL binding buffer and stained with 1 μL APC-Annexin V (BD Biosciences) for 15 min. The apoptosis analysis was performed using flow cytometry (TreeStar, Ashland, OR, USA).

### 2.14. Western Blotting Assay

Proteins were resolved by 12% SDS-polyacrylamide gel electrophoresis under reducing conditions. The resolved proteins were transferred electrophoretically to a polyvinylidene fluoride membrane. After incubating with 5% non-fat milk in tris-buffered saline with tween (TBS-T) (150 mM NaCl, 50 mM Tris HCl (pH 7.4), 0.05% Tween 20) for at least 1 h at room temperature, the membrane was incubated with primary antibodies (Mfn2, Mfn1, Drp1, PINK1 VADC, ERP57, mt-HSP70 and β-Actin (from Abcam, USA) and Flag (from Thermo Fisher Scientific, USA)) for the appropriate time, washed extensively with TBS-T and then incubated with HRP-conjugated secondary antibody (1:2000 dilution). Protein bands were detected using a Super Signal West Pico Chemi-Luminescent Substrate Kit (Pierce, Thermo Fisher Scientific, Rockford, IL, USA). Densitometric analyses were done using Image J software.

### 2.15. Real-Time Quantitative RT-PCR

The cells were diluted to 2 × 10^5^ cells/mL in six-well plates in 5% CO_2_ at 37 °C for 2–3 days to prepare cells achieving 60% confluence. Cells were washed once with PBS, then mixed with 1 mL RNAiso Plus (TaKaRa, 9109) for 5 min and 200 µL chloroform for 10 min. After centrifuging at 14,000 r/min for 20 min at 4 °C, the supernatant was collected and added 400 µL isopropanol for 10 min and repeat centrifuging in the same condition. Then, 1 mL 75% precooled ethanol was put into the precipitate and centrifugation at 7500 r/min for 1 min at 4 °C. After removing the supernatant, we applied 20µL ddH_2_O and then RNA were saved. The reaction mixtures 1 (10 µL, includs 2 µL 5× gDNA Eraser Buffer, 1 µL gDNA Eraser, 1µg Total RNA and RNase Free dH_2_O) stay in 42 °C for 2 min. The reaction mixtures 2 (20 µL, contains 1 µL PrimeScript RT Enzyme Mix I, 1µL RT Primer Mix, 4 µL 5× PrimeScript Buffer 2 (for Real Time), 10 µL reaction mixtures 1 and 4 µL RNase Free dH_2_O) stay in 37 °C for 15 min, 85 °C for 5 s and 4 °C forever. The reaction mixtures 3 were performed in a final volume of 25 µL containing 2 µL reaction mixtures 3 (cDNA), 0.4 µL upstream primer, 0.4 µL downstream primer, 10 µL 2 × plus SYBR real-time mixture (Vazyme, Q311-02/03), 0.4 µL ROX I and 6.8 µL dH_2_O. The quality of the specimens was assessed by a real-time RT-PCR targeting the glyceraldehyde-3-phosphate dehydrogenase (GAPDH) gene. Amplification was performed on a Bio-Rad IQ real-time PCR system (Bio-Rad, Marnes la Coquette, France). Sterile diethyl pyrocarbonatetreated water was used as an RT- and PCR-negative control.

### 2.16. Statistical Analysis

Data in graphs are reported as ± SE and depict the average of at least three independent experiments. Morphological images are representative of at least three independent experiments with similar results. Statistical analysis was performed using GraphPad software (GraphPad Software, La Jolla, CA, USA) with Student’s paired *t*-test, or a one-way analysis of variance followed by a Dunnett’s multiple comparison test when appropriate.

## 3. Results

### 3.1. Changes in Mitochondrial Morphology and the EBV Encoded Protein BHRF1 in NPC

The nasopharyngeal carcinoma (NPC) and nasopharyngitis are two common pathological changes in nasopharynx. The initial clinical symptoms of nasopharyngeal carcinoma such as epistaxis, nasal congestion and stuffy nose are generally consistent with nasopharyngitis. Patients with such symptoms were consulted by doctors in otolaryngology department of the hospital, and were followed with examinations by electronic fiber laryngoscope. The tissues (2 × 2 mm) were obtained from the posterior of parapharyngeal. To directly address the mitochondrial implications in EBV mediated NPC, we collected 41 NPC tissue specimens, and compared them to the tissue of 19 patients with nasopharyngitis with similar age and sex distribution from the same location (Table 1). We used transmission electron microscope (TEM) to observe the mitochondria in these tissues. Compared with those in nasopharyngitis tissue, some mitochondria showed significant swelling in the NPC tissue cells (Figure 1A,B). In addition, the cristae in NPC mitochondria were more heterogeneous. They were not arranged in parallel stacks, and less interconnected to one another and to the inner boundary membrane (Figure 1A).

In order to confirm the infection of EBV in the tissue samples and to explore the involvement of mitochondria in EBV dependent NPC, we measured the expression of several EBV proteins (via quantitative RT-PCR), which have previously been implicated in tumorigenesis and mitochondrial regulation [19]. Among BALF1, BARF1, LMP1, and BHRF1, we found the mRNA level of BHFR1 in NPC tissues was significantly higher than that in nasopharyngitis (all *p* < 0.0001) (Figure 1C). For comparison, we also measured the expressions of such EBV proteins in an EBV positive NPC C666-1 cells. While all EBV encoded proteins in C666-1 cells were expressed at a higher level than what was found in tissues, we achieved to express BHRF1 gene in our CNE1 BHRF1 line higher than that in C666-1 cells (Appendix A). In addition, most EBV-endcoded genes were expressed at comparable levels (Appendix A).

### 3.2. The Expression of BHRF1 in NPC Cells Changes Mitochondrial Morphology

To further explore the potential role of BHRF1 in EBV mediated NPC tumorigenesis, we selected two EBV-negative NPC cell lines with different metastatic properties, CNE1 [20] and 5-8F [21], as the recipients for BHRF1 transfection. Lentiviral expression vectors were used to introduce the BHFR1 gene into CNE1 and 5-8F NPC cells. Multiple transfectants were obtained, and two representatives with different expression levels for each parental line CNE1-BHRF1 (L), CNE1-BHRF1, 5-8F-BHRF1 (L), 5-8F-BHRF1 were picked for further studies (Figure 2A). A qRT-PCR analysis (Table 2) was used to confirm the expression of BHRF1 at the mRNA level, and to compare it with the expression in NPC tissues from patients with EBV infection. It is worth noting that the expression of BHRF1 in all of our selected transfectants was comparable to that of NPC tissues.

TEM was then used to verify that BHRF1 had an effect on mitochondrial features similar to those observed in NPC tumor tissues. As shown in Figure 2B, some of the mitochondria in BHRF1-transfected cells were somewhat swollen. Interestingly the EBV-positive C666-1 exhibited a similar feature as observed in BHRF1 transfected cells (Appendix A). Quantitative analysis of this swelling showed that BHRF1 protein increased mitochondrial volumes (all *p* < 0.05) (Figure 2C).

### 3.3. The Expression of BHRF1 in NPC Cells Enhanced Tumorigenesis

In order to investigate the contribution of BHRF1 to NPC carcinogenesis, we analyzed tumorigenesis features in BHRF1 transfectants and control lines. In a colony formation assay, we found BHRF1-expressing cells formed larger and more clones than controls (Figure 3A). As a complementary approach, we then measured cell growth capacity. The result showed that the proliferation rates of CNE1 and 5-8F cells with BHRF1 protein increased significantly at 24 h and 48 h, respectively (Figure 3B). To study the migration and invasion ability BHRF1 expression conferred to NPC cell lines, a cell wound closure assay was first performed. A wound was introduced into a confluent monolayer of cells to create a cell-free zone into which cells at the edges of the wound can migrate. The images of cell movement were captured after 0, 24, 48, and 72 h of wound formation. As shown in Figure 3C, cell migration was enhanced in the CNE1 BHRF1 and 5-8F BHRF1 cells. We then carried out complementary transwell migration and invasion assays. As shown in Figure 3D,E, the BHRF1 expression significantly elevated the number of migration cells that penetrated the transwell, as well the number of invasion cells that penetrated the matrigel.

### 3.4. Localization of BHRF1 to Mitochondria in NPC Cells and the Implications of BHRF1 Activity on Mitochondria Function

BHRF1 has been previously suggested to localize to mitochondria [9,22]. To confirm and characterize its localization in NPC cells, we introduced PCDH-BHRF1-flag plasmid into the above NPC cells. The whole cell lysate (W), mitochondria-enriched precipitate fraction (M), mitochondrial matrix-enriched supernatant fraction (M1), and mitochondrial membrane-enriched precipitate fraction (M2) were obtained through differential centrifugations as described in the methods. The success of the fractionation was validated by detection of various markers such as endoplasmic reticulum protein 57 (ERP57) for ER, VADC for mitochondrial outer membrane, and mt-HSP70 for mitochondrial matrix. An anti-Flag antibody was used to determine BHRF1 protein localization. We found BHRF1 indeed localized to the mitochondria, and more specifically it was found in the mitochondrial membrane fraction (M2) (Figure 4A).

To determine the effects of BHRF1 on mitochondrial function and the implications of mitochondrial swelling in NPC cells, we measured the mitochondrial membrane potential (MMP) with TMRE. As shown in Figure 4B, the MMP was significantly decreased with the expression of BHRF1 in both CNE1 and 5-8F cells. We then analyzed the ATP production, and found that BHRF1 expression compromised ATP generation in CNE1 (*p* < 0.005) and 5-8F cells (*p* < 0.05) (Figure 4C). To determine if there is an alteration in ROS production associated with the MMP and ATP reduction, the mitochondria-specific generation of superoxide was analyzed by MitoSOX. We found an increase in mitochondrial ROS production with the BHRF1 expression in CNE1 and 5-8F cells (Figure 4D). To investigate whether the decrease in energy production and increase in oxidative stress would change the cell viability, flow cytometry was used to quantify the amount of apoptotic cells detected by Annexin V (Annexin V-FITC) and 7-aminoactinomycin D (7-AAD). Quantitative descriptive analysis of flow cytometry showed that with BHRF1 expression in NPC cell lines, the apoptosis rate was significantly decreased (all *p* < 0.05) (Figure 4E).

### 3.5. BHRF1 Regulation of ROS Production and Cell Survival is Mediated through MMPT

To further investigate the underlying mechanisms of putative BHFR1 mediated NPC tumorigenesis with compromised oxidative phosphorylation, increased ROS production and increased cell survival, we decided to look into the role of the mitochondrial membrane permeability transition (MMPT). This could be a promising new link in the mechanism, as mitochondrial swelling and compromised MMP have been associated with the MMPT, which is also considered as the critical stage for cell death/survival. Cyclosporine A (CsA) is a potent inhibitor for the MMPT, binding to mitochondrial cyclophilin to prevent the opening of mitochondrial permeability transition pores [23]. As shown in Figure 5A, CsA treatment reversed the reduction in MMP caused by the expression of BHRF1 in both NPC cell lines. Similarly, CsA treatment also reversed the decrease in ATP production (Figure 5B) and increase in ROS generation (Figure 5C) associated with BHRF1 expression in CNE1 and 5-8F cells. The apoptosis rate as demonstrated by flow cytometry also increased significantly at the optimal concentration of CsA treatment with CNE1 and 5-8F cells (Figure 5D).

### 3.6. The Effect of MMPT-Mediated BHRF1 Regulation on Tumorigenesis

To further explore how the MMPT induced by BHRF1 may mediate tumorigenesis, we repeated the tumorigenesis assays performed in the previous sections in presence of the MMPT inhibitor CsA. In a colony formation assay, the number and volume of colonies all decreased after CsA treatment in NPC cells with BHRF1 expression (Figure 6A). The CCK-8 cell viability assay (Figure 6B), the wound closure assay (Figure 6C), transwell migration (Figure 6D), and invasion (Figure 6E) assays all showed that the enhanced tumorigenesis features conferred by BHRF1 were reversed by the inhibition of the MMPT achieved by CsA treatment.

### 3.7. Mitophagy Activation in MMPT Dependent Tumorigenesis in NPC Cells and Tissues

To determine the molecular mechanism by which the BHRF1-reguated MMPT modulates NPC tumorigenesis, we analyzed the protein levels by Western blot of critical factors that have the potential to mediate the retrograde signaling from mitochondrial alteration to tumorigenesis. Among them, we found multiple proteins related to mitochondrial dynamics and mitophagy exhibited significant changes. As shown in Figure 7A, while mitochondrial fusion marker Mfn1 and Mfn2 levels decreased with the expression of BHRF1 in NPC cells, both mitochondrial fission (Drp1) and mitophagy (PINK1) related proteins increased. Therefore, we focused on the effect of autophagy in this process. We analyzed autophagy by detection of LC3 protein process. As LC3-II correlates well with the number of autophagosomes, it provides a good index for autophagy induction. As shown in Figure 7B, the LC3II/LC3I ratio significantly increased with BHRF1 expression. Such induction, however, was reversed with MMPT inhibitor CsA treatment (Figure 7C). Furthermore, the enhanced induction of autophagy indicated by the LC3 protein processing was also found in the NPC tumor tissues when compared with the matched nasopharyngitis tissues (Figure 7D).

## 4. Discussion

EBV was the first human virus to be directly implicated in tumorigenesis, and has been linked to the development of cancers originating from both lymphoid and epithelial cells [24]. Interestingly although more than 90% of world’s population sustains a life-long infection with EBV, only small portion of the population will eventually develop tumors [25]. The role of EBV in the carcinogenesis of NPC is currently poorly understood, but it is widely believed that the presence of the virus in all tumor cells provides opportunities for developing therapeutic and diagnostic approaches.

EBV encodes nearly a hundred genes, and among them the EBV latent genes expressed in NPC are clearly essential in the carcinogenic process. While the EBNA proteins have been shown to be important in replication and segregation of the EBV genome and viral gene expression, LMP proteins have been shown to play various roles contributing to tumorigenic phenotypes [26,27]. EBV also encodes several important proteins that show sequence homology to diverse human proteins including BCRF1 for IL-10, BDLF2 for Cyclin B1, BARF-1 for Intracellular Adhesion Molecule 1, and BHRF1 for Bcl-2 [4].

In an effort to investigate the role of mitochondria in EBV dependent NPC tumorigenesis, we explored the abnormal mitochondrial morphology (Figure 1A) associated with high expression of BHRF1 (Figure 1C) in NPC tumor tissues. Altered expression of Bcl-2 genes has been associated with a number of cancers, including melanoma, breast, prostate, chronic lymphocytic leukemia, and lung cancer [28]. Bcl-2 family proteins have been shown to play an important role in promoting cellular survival as anti-apoptotic proteins, or pro-apoptotic proteins (including Bax and Bak) through the release of cytochrome C and other apoptosis inducing factors [29]. BHRF1 exhibits homology to the Bcl-2 family proteins Bcl-x (32%) and Bax (34%) over the carboxyl portion, with the same BH1 and BH2 domains [30], while it lacks the prominent hydrophobic groove that mediates binding to pro-apoptotic family members Bak, Bax, Bik, and Bad [31,32].

The expression of BHFR1 in two NPC cell lines reproduced our findings in the NPC tumor tissues that showed changes in the mitochondrial morphology (Figure 2). Further, with the expression of BHRF1, tumorigenesis features are all enhanced as displayed by colony formation, wound closure, cell migration, and invasion assays (Figure 3). Like Bcl-2, BHRF1 also localized to the mitochondria (Figure 4A), and BHRF1 compromised the MMP associated with alterations in ROS production and apoptotic properties (Figure 4B,D,E).

Apoptosis, mainly regulated by mitochondria, is one of the important regulatory pathways in tumorigenesis. Mechanistically, MMPT associated with an increase in the permeability of the mitochondrial membrane leads mitochondrial swelling and rupture of the outer mitochondrial membrane. This has been demonstrated to mediate the depolarization of the transmembrane potential as illustrated by the reduction of the MMP, release of apoptotic factors, and decrease of oxidative phosphorylation [33,34]. Thus MMPT is considered a critical step of the apoptotic cascade [35]. Moreover, ischemia-damaged mitochondria with decreased Bcl-2 content are susceptible to MPTP opening in early reperfusion and mitochondrial outer membrane permeabilization (MOMP) later in reperfusion when cytosolic calcium has normalized [36].

One interesting finding of this study is that the CypD dependent MMPT is regulated by Bcl-2 homolog BHRF1. The coupling of transitions of out mitochondrial membrane (OMM) and inner mitochondrial membrane (IMM) has previously been shown that the activation of the permeability transition pore (PTP) at least in vitro, resulted in passive swelling of the IMM and rupture of the OMM, with ensuing release of IMS components. In particular, over expression of Bcl-2 increased the resistance of IMM transition [37]. In addition, a co-immunoprecipitation study showed the direct interaction between cyclophilin D and Bcl-2 [38]. CsA, as a specific inhibitor of the cyclophilin family, specifically blocked the MMPT [39]. With CsA treatment, we successfully reversed the mitochondrial alteration, ROS generation and increased apoptotic properties induced by high expression of BHRF1 in both NPC cells (Figure 5). Further, this MMPT inhibitor also reversed the tumorigenesis and metastatic features associated with BHRF1 expression. Our results showed that mitochondrial parameters affected by BHRF1 are inhibited by Cyclosporin A, and when combined with findings from the aforementioned studies these results suggest that CypD dependent MPTP is the key mechanism in BHRF1-mediated tumorigenesis in NPC cells.

The MMPT has been considered to be a ‘point of no return’ during apoptosis, which once was suggested to inevitably lead to cell death. One key event after the MMPT is the triggering of autophagy [40]. Quite a few regulatory factors are also released from mitochondria following the MMPT. For example, Smac/Diablo promotes caspase activation through neutralizing inhibitor of apoptosis proteins (IAPs) [41]. In this study, we provided evidence that under certain circumstances, MMPT could also activate mitophagy as a cell survival mechanism.

Alterations in the autophagy pathway have been associated with the formation of various cancers. Autophagy is also an adaptive catabolic process that happens in response to metabolic stress, including hypoxia and ROS. One possible mechanism here is that NPC exploits the enhancement of autophagy for survival to adapt to the hypoxia of a tumor microenvironment [42,43]. Our study suggests that the BHRF1 mediated MMPT can also utilize mitophagy as a cell survival mechanism during EBV dependent NPC tumorigenesis. Interestingly, previous research showed that decreasing the activity of Drp1, a protein involved in mitophagy that is mainly responsible for mitochondrial division, could enhance the treatment of NPC [44].

It is also important to note that our studies don’t necessarily exclude the contributions of other EBV encoded protein in NPC tumorigenesis. Rather, our results describe a scenario in which EBV encoded BHRF1 induces MMPT, and such MMPT further regulates NPC tumorigenesis through ROS production and mitophagy activation (as shown in the graphic abstract).

## Figures and Tables

**Figure 1 cells-09-01158-f001:**
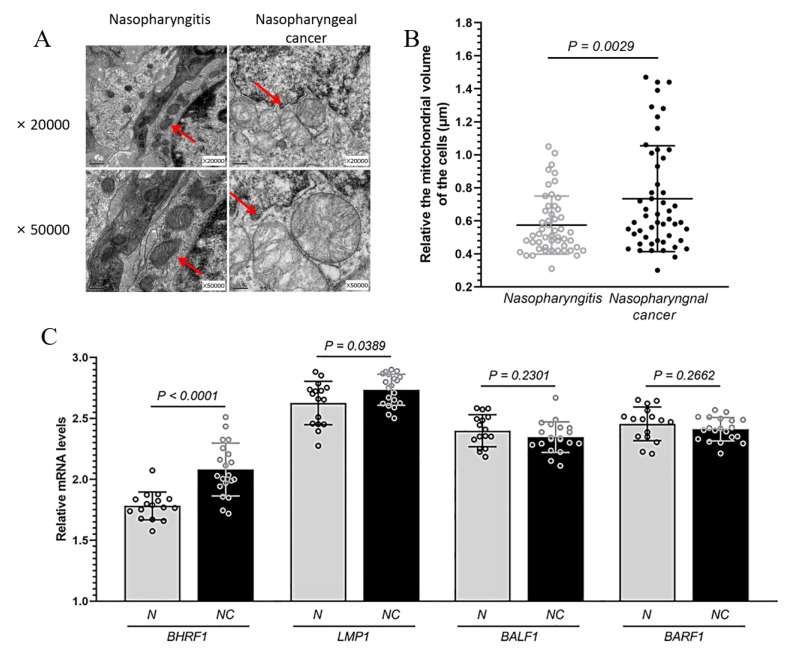
The changes in mitochondrial morphology and EBV encoded proteins in NPC. Nasopharyngeal carcinoma (NPC) and nasopharyngitis are two common pathological changes in nasopharynx. (**A**), Mitochondrial morphology analyzed under transmission electron microscopy in the tissue of patients with NPC and nasopharyngitis at ×20,000 and ×50,000. Mitochondria are indicated by arrows. (**B**), The relative volume of about 50 mitochondria each in the tissue of patients with NPC and nasopharyngitis. Mitochondrial size was measured using Image J by manually tracing outlines of mitochondria on TEM micrographs. (**C**), Levels of BHRF1, LMP1, BALF1 and BARF1 mRNA expression from 20 NPC (NC) and 16 nasopharyngitis (N) patients detected via qRT-PCR.

**Figure 2 cells-09-01158-f002:**
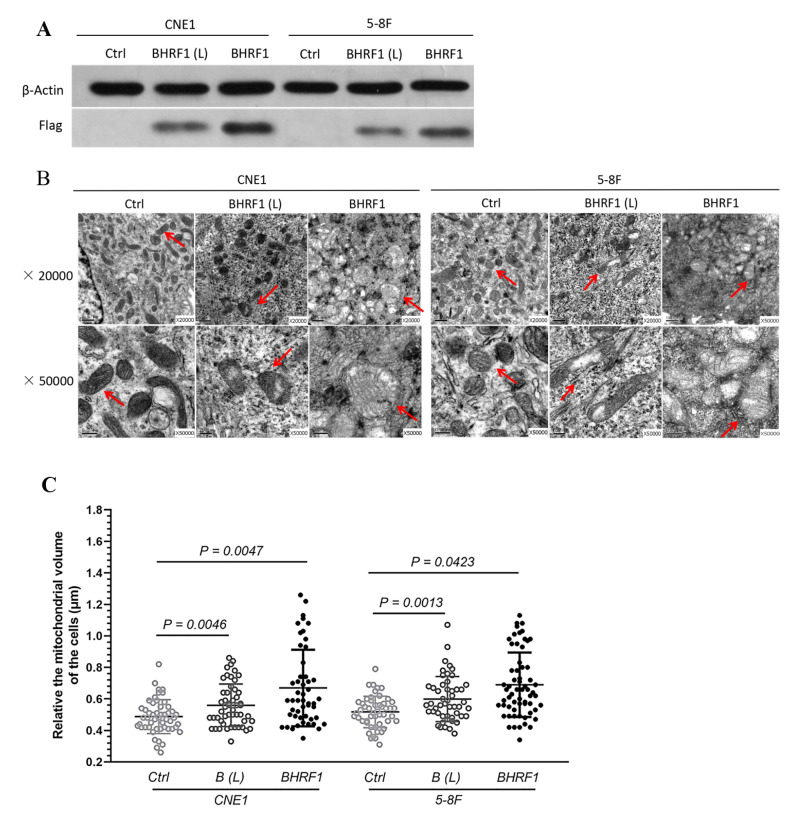
The high expression of BHRF1 in NPC cells alters mitochondrial morphology. (**A**), Western blot analysis of BHRF1 expression from the cell lysates of independent clones of CNE1 (CNE1 BHRF1(L, for low expression) and CNE1 BHRF1) and 5-8F (5-8F BHRF1(L) and 5-8F BHRF1) cells using anti-Flag antibody (1:1000 dilution). (**B**), Mitochondria in NPC cells observed under transmission electron microscopy. (**C**), Fifty mitochondria were randomly selected from three images to each cell line to measure their sizes. The *p* value was evaluated by a paired *t*-test. Image J was uesd to measure mitochondria area on TEM micrographs.

**Figure 3 cells-09-01158-f003:**
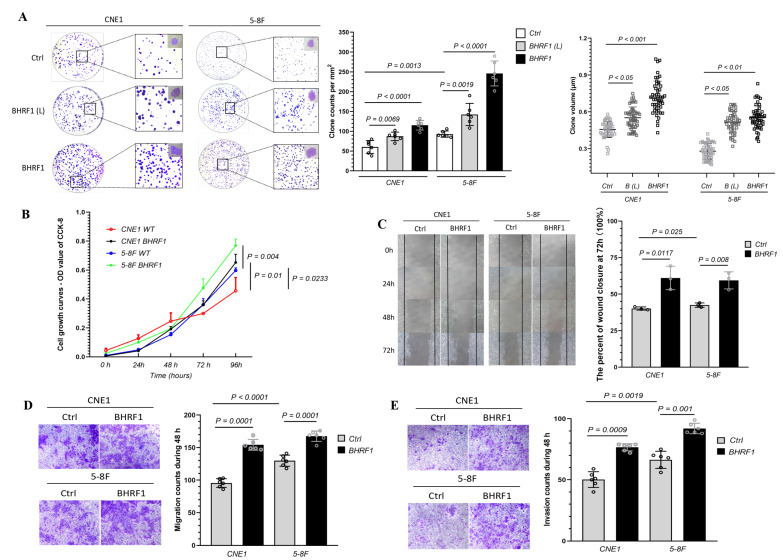
The high expression of BHRF1 enhanced tumorigenesis in NPC cells. (**A**), An equal number of cells (four replicates for each line) were suspended in DMEM medium with 10% FBS and seeded on 60 mm dish. The colonies were analyzed after 3 weeks and representative pictures of one of the three experiments were shown. The colonies 0.1 mm were scored with alpha imager colony-counting software. (**B**), Cellular viability after 0 h, 24 h, 48 h, 72 h, 96 h of growth in DMEM media was quantitated using Vi-Cell cell viability analyzer based on the trypan blue exclusion method. (**C**), Images of cell movement were captured at 0, 24, 48, 72 h and used to calculate the rate of wound closure by ImageJ. (**D**), Cell migration was analyzed using trans-well assay (four wells/line) and cells that migrated to the lower surface were stained and imaged. E, Cell invasion was analyzed by the matrigel solution and trans-well. A total of three independent fields per well were selected for counting invasion cells using NIS-e lement software. Representative images (*n* = 3) are shown and quantitation represents mean ± SE of three independent experiments.

**Figure 4 cells-09-01158-f004:**
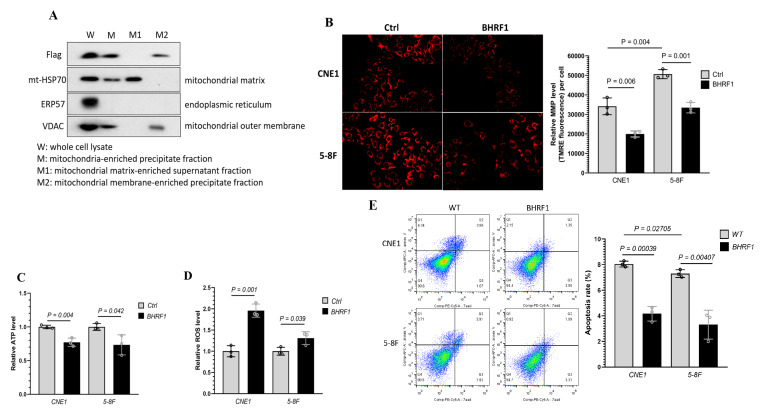
Mitochondrial localization of BHRF1 protein and the implications of BHRF1 on mitochondria function. (**A**), Western blots were used to detect BHRF1, VADC, ERP57 and mt-HSP70 proteins using specific antibodies for each protein, with the exception of BHRF1, which was detected with an anti-Flag antibody. (**B**), MMP was determined with TMRM fluorescence in BHRF1 high expression cells and control one. (**C**), Total ATP content was determined in BHRF1 high expression NPC cells and control ones (*n* = 3). (**D**), Mitochondrial ROS levels were determined in BHRF1 high expression NPC cells and control ones (*n* = 3). The levels of ATP and ROS were normalized via protein concentration. (**E**), Cell apoptosis rate was measured by FCM (flow cytometry). Representative images are shown and quantitation represents mean ± SE of three independent experiments.

**Figure 5 cells-09-01158-f005:**
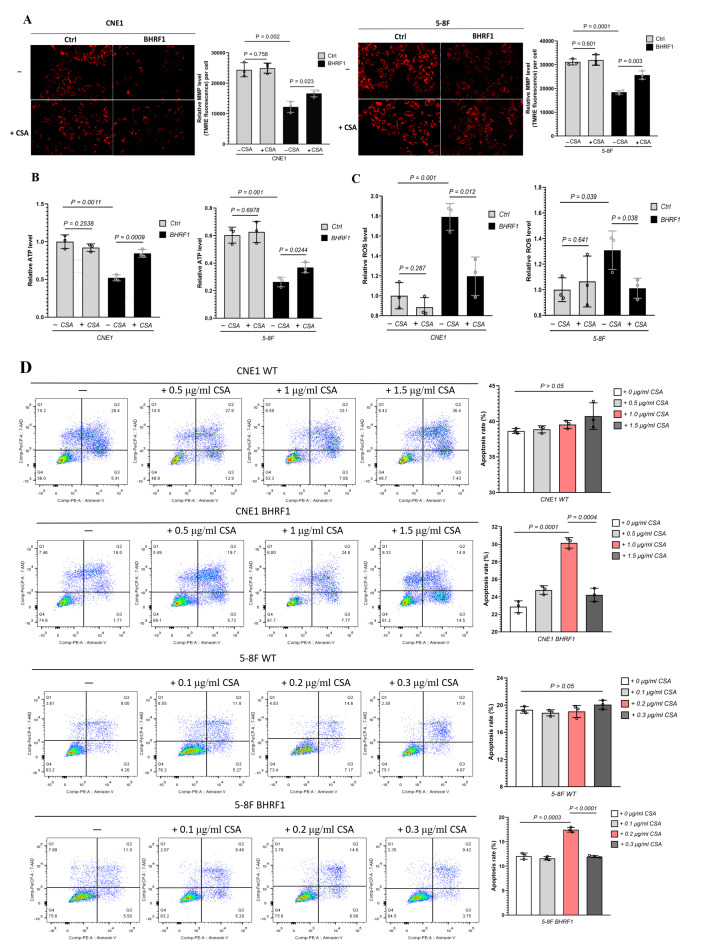
BHRF1 regulation is mediated through MMPT. (**A**), Representative images for assessment of MMP by TMRE for NPC cell lines treated with CSA; (**B**) and (**C**), ATP production or ROS generation in control cells and BHRF1 high expression cells treated with CsA. (**D**), The apoptosis rate of NPC cell lines with gradient concentration of CsA by flow cytometry. Data represent mean ± SE of three independent experiments.

**Figure 6 cells-09-01158-f006:**
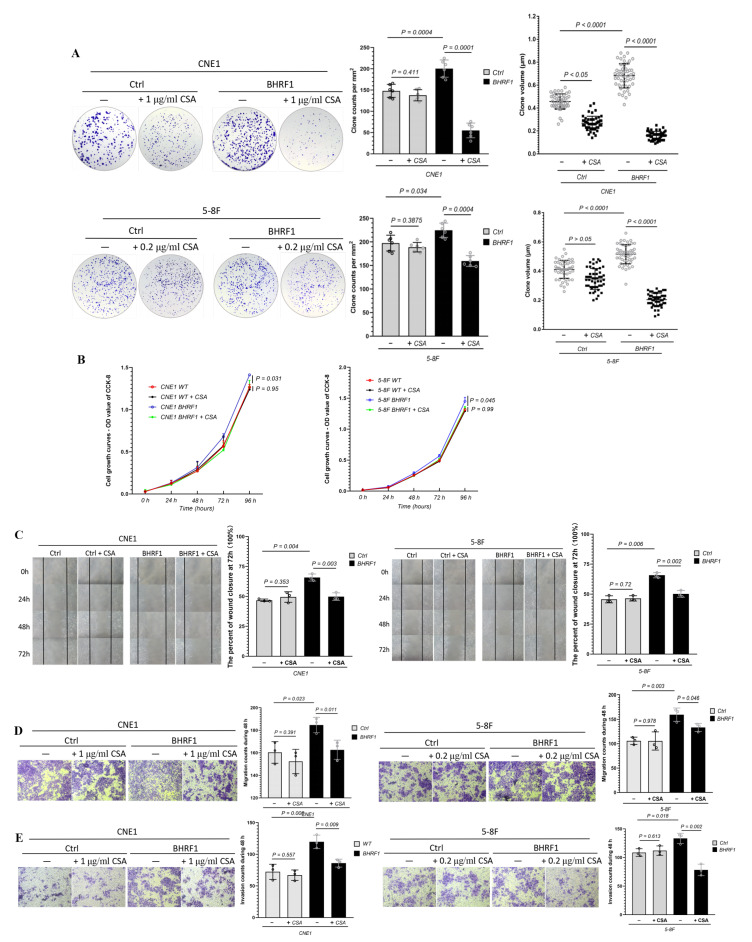
MMPT mediates BHRF1-induced tumorigenesis. (**A**), Colony formation assay of BHRF1 high expression cell lines and the controls with CsA; (**B**), The proliferation of WT and BHRF1 cells with CSA with CCK-8 assay. (**C**), Wound closure assay with CsA; (**D**), Transwell migration assay with CsA and (**E**), Transwell invasion experiment in CNE1, CNE1 BHRF1 and 5-8F, 5-8F BHRF1 cells with CsA. Representative images (*n* = 3) are shown and quantitation represents mean ± SE of three independent experiments.

**Figure 7 cells-09-01158-f007:**
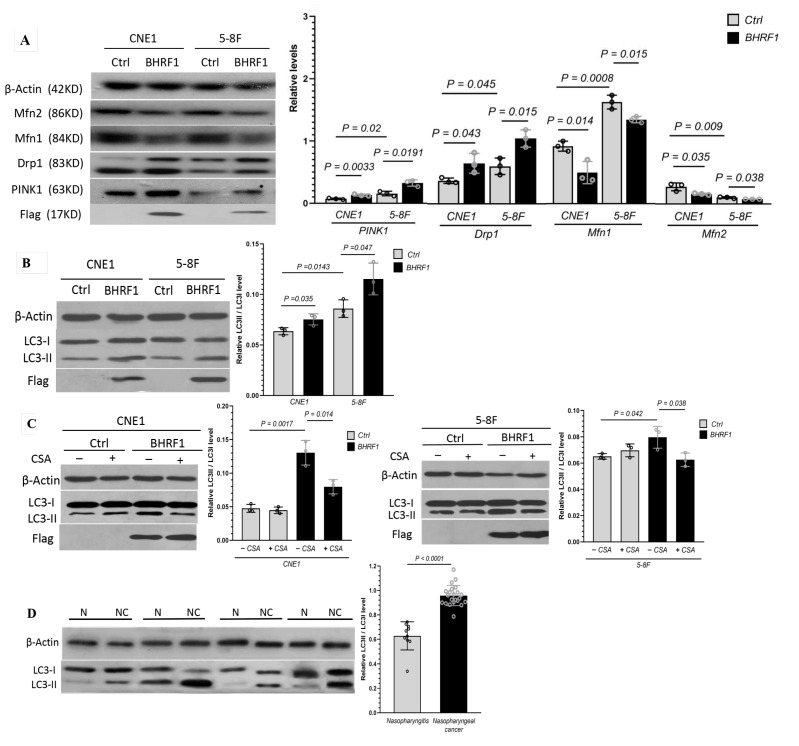
Mitophagy mediates the effect of MMPT on cell survival in NPC cells and tissues. (**A**), Relative protein expression of Mfn2, Mfn1, Drp1, PINK1 in control cells and BHRF1 high expression cells. (**B**), The LC3II/LC3I protein level in NPC cell lines by western blotting; (**C**), The LC3II/LC3I protein level with or without CsA by western blotting in NPC cell lines; (**D**), The expressions of LC3II/LC3I protein level in nasopharyngeal cancer and nasopharyngitis tissues by WB.

**Table 1 cells-09-01158-t001:** Patient demographics of NPC and nasopharyngitis patients.

Clinicopathological Features	Nasopharyngitis	Nasopharyngeal Cancer	*p*
Cases	19	41	
Age (years old)			
Average	52	51	***p*** > 0.05
S. D.	11.3	11.6	
Range	30–73	23–74	
Gender			***p*** > 0.05
Female	10	10	
Male	9	31	

**Notes:** The value of age groups was evaluated by an independent sample-test, simultaneously the P value of gender groups was evaluated by Fisher’s exact test.

**Table 2 cells-09-01158-t002:** Relative BHRF1 mRNA expression between high BHRF1 expression NPC cell lines and patient tissues.

Tissues/Cell Lines	Relative Expression of BHRF1 mRNA
Nasopharyngitis	1.71 ± 0.26
Nasopharyngitis cancer	2.09 ± 0.39
CNE1 BHRF1 (L)	1.76 ± 0.09
CNE1 BHRF1	1.96 ± 0.05
5-8F BHRF1 (L)	1.71 ± 0.07
5-8F BHRF1	1.94 ± 0.09

**Notes:** BHRF1, BamHIfragment H rightward open reading frame 1 of EBV.

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
