# Peer review of "BHRF1 Enhances EBV Mediated Nasopharyngeal Carcinoma Tumorigenesis through Modulating Mitophagy Associated with Mitochondrial Membrane Permeabilization Transition"

_cells, 2020, doi:10.3390/cells9051158_

Round 1

Reviewer 1 Report

Song et al. have now provided an improved manuscript with major correction compared to their earlier version. However, this reviewer has certain concerns that must be addresed prior to manuscript acceptance. The main concern are the figures 2-7, wherein the stats of the respective graphs are unreadable and too small. Also, this reviewer must insist the  authors to show transcript level of LMP2 and EBNA1. The expression of these proteins are independent of LMP1.

Author Response

This is the revised manuscript by Song et al. The manuscript is greatly improved, and authors clarified all my previous concerns. However, I still have two more minor comments. 

Response: We would like to thank the reviewer for the appreciation of our revised manuscript. 

  1. In all the TMRE measurements, how normalization is performed in the quantification of the TMRE? The TMRE florescence should be normalized to the number of cells.

Response: The TMRM fluorescence was normalized with protein concentration (mg/ml). Per reviewer’s suggestion, we now normalized results to the number of cells. The figures are updated.

  1. Figure 4A: Authors performed a mitochondrial fractionation to validate the mitochondrial localization of BHRF1 protein. Please rename the GRP75 to mt-HSP70 because GRP75 is another protein present at the ER-mitochondrial membrane, and this name is confusing.

Response: We have made the changes throughout the whole manuscript accordingly.

Reviewer 2 Report

This is the revised manuscript by Song et al. The manuscript is greatly improved, and authors clarified all my previous concerns. However, I still have two more minor comments.  

1. In all the TMRE measurements, how normalization is performed in the quantification of the TMRE? The TMRE florescence should be normalized to the number of cells.

2. Figure 4A: Authors performed a mitochondrial fractionation to validate the mitochondrial localization of BHRF1 protein.  Please rename the GRP75 to mt-HSP70 because GRP75 is another protein present at the ER-mitochondrial membrane, and this name is confusing.  

Author Response

Song et al. have now provided an improved manuscript with major correction compared to their earlier version.

Response: We would like to thank the reviewer for the appreciation of our revised manuscript. 

  1. However, this reviewer has certain concerns that must be addresed prior to manuscript acceptance. The main concern are the figures 2-7, wherein the stats of the respective graphs are unreadable and too small.

Response: We now adjusted the resolution and the font size in all figures to make sure they are readable.  

  1. Also, this reviewer must insist the  authors to show transcript level of LMP2 and EBNA1. The expression of these proteins are independent of LMP1.

Response: We really appreciate the reviewer’s suggestions, and would very much like to accommodate. However, due to the recent pandemics, we unfortunately lost the all the tissues samples. Nevertheless, we did measure the transcript level of LMP2A and EBNA1 in the EVB positive C666 cells, and the results are shown in Figure Supplementary 1B.

Reviewer 3 Report

Song and co-workers here present a study where the EBV protein BHRF1 appears to localize to the mitochondria to cause mitochondrial morphological as well as functional changes.

The authors have performed a applaudable amount of work here. The writing is generally clear, and results are sound. I have some comments here, and while additional experiments will always be helpful, I understand given the current COVID situation, it may not be possible. I am ok with the authors addressing these comments with either rewording of better explanations in the manuscript.

1. The methods for qRT is missing, with no explanation for the controls etc. Using the right normalization methods as well as target (actin etc) is very important, particularly since the viral genome can be variable in copy number.

2. How were the TEM mitochondria size measured?

3. In Figure 4, while the fractionation is a classical assay, why is IF for colocalization of BHRF1 and mitochondria not performed?

4. How are the image intensities normalized for Figure 4B? There can be artefacts due to different sample preparations and staining efficiencies.

5. The image quality for Figure 7A is unclear, particularly for the quantification.

6. The authors can discuss why were autophagosome images not show, given the amount of work dedicated to showing their presence via western blots? Typically LC3II/LC3I imaging is performed and autophagosome numbers quantified.

7. The authors discuss a number of details regarding Caspase activity and its relationship with BHRF1. It would be good to show some of these, or include a figure with the mechanism proposed for how BHRF1 induces caspase activity through mitochondrial functional changes.

Author Response

The authors have performed a applaudable amount of work here. The writing is generally clear, and results are sound. I have some comments here, and while additional experiments will always be helpful, I understand given the current COVID situation, it may not be possible. I am ok with the authors addressing these comments with either rewording of better explanations in the manuscript.

Response: We would like to thank the reviewer for the appreciation of our work as “ an applaudable amount of work” and “writing is generally clear, and results are sound”. In particular, we are grateful that the reviewer’s understanding the current difficulties, and is willing to be “ok with the authors addressing these comments with either rewording of better explanations in the manuscript.”

  1. The methods for qRT is missing, with no explanation for the controls etc. Using the right normalization methods as well as target (actin etc) is very important, particularly since the viral genome can be variable in copy number.

Response:  Thank you for the nice suggestion. We now added a new section “2.15 Real-time quantitative RT PCR” in methods part. We used the standard normalization method with the house keeping gene glyceraldehyde-3-phosphate dehydrogenase (GAPDH) as a control.

  1. How were the TEM mitochondria size measured?

Response: Mitochondrial size was measured using Image J by manually tracing outlines of mitochondria on TEM micrographs.

  1. In Figure 4, while the fractionation is a classical assay, why is IF for colocalization of BHRF1 and mitochondria not performed?

Response: Unfortunately with the antibody against Flag we used to detect BHRF1, it works well for Western but not for IHC. We had to resort to the traditional fractionation method for BHRF1 localization.

  1. How are the image intensities normalized for Figure 4B? There can be artefacts due to different sample preparations and staining efficiencies.

Response: First we were careful to make sure two cell lines were maintained and treated in the same condition. We also treated the cells with FCCP at end of experiment and subtracted off remaining fluorescence, which was non-specific FCCP would provide readings after membrane potential was collapsed. To determine the membrane potential, we used fluorescence ratio (F-F0)/F0 to represent the potential, where F0 is the baseline fluorescence signal. We added this clarification in the revised text.

  1. The image quality for Figure 7A is unclear, particularly for the quantification.

Response: We now replaced the original image with one with high resolution.

  1. The authors can discuss why were autophagosome images not show, given the amount of work dedicated to showing their presence via western blots? Typically LC3II/LC3I imaging is performed and autophagosome numbers quantified.

Response: We certainly appreciate your suggestion. In fact, we tried multiple times to image autophagosomes, but all these efforts failed in detection of LC3. It could be due to the malfunction of our imaging core. We are really sorry for this.

7.The authors discuss a number of details regarding Caspase activity and its relationship with BHRF1. It would be good to show some of these, or include a figure with the mechanism proposed for how BHRF1 induces caspase activity through mitochondrial functional changes.

Response: Since this work focused on the activation of autophagy resulting from the alteration of MMPT, and we didn’t analyze the caspase activities in our work. We now instead deleted the related discussion. Nevertheless, the reviewer’s point is well-taken, and we now provide a graphic abstract proposing the working mechanism which links the BHRF1 induces autophagy activation through mitochondrial function changes.